# The Adaptive Role of Carotenoids and Anthocyanins in *Solanum lycopersicum* Pigment Mutants under High Irradiance

**DOI:** 10.3390/cells12212569

**Published:** 2023-11-03

**Authors:** Aleksandr Ashikhmin, Maksim Bolshakov, Pavel Pashkovskiy, Mikhail Vereshchagin, Alexandra Khudyakova, Galina Shirshikova, Anna Kozhevnikova, Anatoliy Kosobryukhov, Vladimir Kreslavski, Vladimir Kuznetsov, Suleyman I. Allakhverdiev

**Affiliations:** 1Institute of Basic Biological Problems, Russian Academy of Sciences, Pushchino 142290, Russia; ashikhminaa@gmail.com (A.A.); lfbv22@gmail.com (M.B.); s_t_i_m_a_@mail.ru (A.K.); gsh99@rambler.ru (G.S.); kosobr@rambler.ru (A.K.); vkreslav@rambler.ru (V.K.); 2K.A. Timiryazev Institute of Plant Physiology, Russian Academy of Sciences, Botanicheskaya Street 35, Moscow 127276, Russia; pashkovskiy.pavel@gmail.com (P.P.); mhlvrh@mail.ru (M.V.); kozhevnikova.anna@gmail.com (A.K.); vlkuzn@mail.ru (V.K.)

**Keywords:** adaptation, *Solanum lycopersicum*, high-intensity light, mutants, photoinhibition, pigments, photosynthetic activity

## Abstract

The effects of high-intensity light on the pigment content, photosynthetic rate, and fluorescence parameters of photosystem II in high-pigment tomato mutants (*hp* 3005) and low-pigment mutants (*lp* 3617) were investigated. This study also evaluated the dry weight percentage of low molecular weight antioxidant capacity, expression patterns of some photoreceptor-regulated genes, and structural aspects of leaf mesophyll cells. The 3005 mutant displayed increased levels of photosynthetic pigments and anthocyanins, whereas the 3617 mutant demonstrated a heightened content of ultraviolet-absorbing pigments. The photosynthetic rate, photosystem II activity, antioxidant capacity, and carotenoid content were most pronounced in the high-pigment mutant after 72 h exposure to intense light. This mutant also exhibited an increase in leaf thickness and water content when exposed to high-intensity light, suggesting superior physiological adaptability and reduced photoinhibition. Our findings indicate that the enhanced adaptability of the high-pigment mutant might be attributed to increased flavonoid and carotenoid contents, leading to augmented expression of key genes associated with pigment synthesis and light regulation.

## 1. Introduction

Photosynthesis involves the absorption of light by chlorophyll and carotenoids. Then, this light energy is converted into chemical energy through photosynthesis. However, when the absorbed light energy exceeds the capacity of the plant to utilize it in photosynthesis, the excess energy can cause damage to the photosynthetic apparatus (PA), especially to photosystem II (PSII), a process known as photoinhibition that occurs in plants when they are exposed to excessive light energy, particularly under stressful conditions such as cold, drought, and nutrient deficiency [1]. Photoinhibition is a dynamic process and can be classified into two types based on its reversibility [2,3]. Dynamic photoinhibition or reversible downregulation of photosynthetic activity is a reversible process that occurs as a protective mechanism in response to temporary changes in light conditions [4]. It involves nonphotochemical quenching (NPQ) mechanisms that dissipate excess light energy as heat, preventing damage to the PA. Another form of photoinhibition is chronic photoinhibition, which is a more long-term and irreversible form of photoinhibition that occurs when plants are exposed to sustained high-intensity light (HIL) [5]. The damage to PSII is significant and requires the synthesis of new proteins for recovery [6,7]. Photoinhibition can significantly affect the overall health and productivity of plants, as it reduces their photosynthetic efficiency. This can result in reduced plant growth and yield. Exposure of plants to HIL can lead to overexcitation of their PA and the formation of ROS such as singlet oxygen (^1^O_2_), superoxide (O_2_⁻), hydrogen peroxide (H_2_O_2_), and hydroxyl radical (·OH) [8,9]. Plants have evolved various strategies to minimize the damage caused by photoinhibition, such as the ability to repair damaged PSII complexes, changes in leaf orientation, and synthesis of protective pigments.

One of the protection mechanisms is nonphotochemical quenching (NPQ), which plays a key role in protecting plants from the adverse action of HIL [10,11]. The process of NPQ involves various components within the chloroplasts, including the light-harvesting complex (LHC), xanthophyll cycle pigments (such as violaxanthin, antheraxanthin, and zeaxanthin), and specific proteins such as PsbS [12]. It consists of the conversion of violaxanthin to zeaxanthin, and vice versa, in response to light and actively functions in conditions of excess light energy [13]. This allows the plant to efficiently use the available light for photosynthesis again [14].

Another important protection mechanism against photoinhibition induced by HIL is a shift in the antioxidant/pro-oxidant balance in the direction of antioxidants, which are a crucial part of the plant defense system against oxidative stress caused by ROS. Plants have a complex antioxidant system to neutralize these harmful ROS [8,9]. This system consists of both nonenzymatic and enzymatic antioxidants, such as superoxide dismutase (SOD), catalase (CAT), and various types of peroxidases, including ascorbate peroxidase (APX) and glutathione peroxidase (GPX) [8,15]. Nonenzymatic antioxidants include compounds such as ascorbic acid (vitamin C), glutathione, carotenoids, and flavonoids. This antioxidant system works in a coordinated manner to maintain the balance between the production and scavenging of ROS, thereby protecting plant cells from oxidative damage. Leaf pigments, as cell antioxidants or optical filters absorbing excess energy, play an important role in mechanisms of protection from HIL [16,17].

Phenolic compounds play a vital role in the plant’s response to various environmental stresses, including HIL. This group encompasses flavonoids, phenylpropanoids, phenolic acids, tannins, and lignins, which collectively contribute to plant protection. In particular, anthocyanins, a subset of phenolic compounds, have the ability to absorb HIL, thus protecting photosynthetic tissues from light-induced damage [18]. They effectively screen light and prevent ROS formation due to excessive light energy. The synthesis and accumulation of phenolic compounds is tightly regulated by plant genetic and metabolic networks [19]. Among the metabolites, phenylalanine ammonia-lyase (PAL) is one of the first enzymes in the phenylpropanoid pathway and is responsible for the biosynthesis of various phenolic compounds [20]. Chalcone synthase (CHS) catalyzes the first committed step in the flavonoid biosynthetic pathway, a subset of the phenylpropanoid pathway [21]. Some members of the bZIP family of transcription factors, such as HY5, can regulate the expression of genes in the phenylpropanoid pathway in response to light exposure [22]. HY5 (elongated hypocotyl 5) is a key transcription factor that promotes photomorphogenesis in light acting downstream of the light receptor network, in spite of the transcription of many light-induced genes [23]. Moreover, phytochrome photoreceptors activate photosynthetic pigment production. DET, a negative regulator of chlorophyll and carotenoid biosynthesis, while HY5, is a potent PIF antagonist that promotes photosynthetic pigment accumulation in response to light. [24]. Anthocyanidin synthase (ANS) is involved in the last steps of anthocyanidin biosynthesis. Anthocyanins are related to flavonoids and are red, blue, and purple pigments in plants with antioxidant properties, and the *ANS* gene is capable of activation by HY5 [25]. The FLS (flavonol synthase) enzyme catalyzes the formation of flavonols from dihydroflavonols and light can stimulate *FLS* gene expression, leading to flavonol production which can protect against UV and HIL damage [26].

Pigment concentration adjustments equip plants to adapt to unfavourable conditions. For example, high chlorophyll content can enhance photon absorption and facilitate an optimal photosynthetic rate, ensuring maximum carbon assimilation [27]. Conversely, a low chlorophyll content causes decreased photosynthetic efficiency, leading to less susceptibility of PA to photodamage under HIL conditions.

Carotenoids play a pivotal role in photoprotective processes. They function as accessory pigments, facilitating energy transfer to photosynthetic reaction centers [28]. More critically, they can dissipate excess excitation energy, especially preventing chlorophyll triplet formation and subsequent ROS generation. Low carotenoid concentrations may be rendered more vulnerable to photodamage owing to a diminished capacity to absorb excess excitation energy, leading to increased oxidative stress [29]. PSY (phytoene synthase) PSY is a key enzyme in the carotenoid biosynthesis pathway. Light can induce *PSY* gene expression, leading to carotenoid production which plays a role in photosynthesis and photoprotection [30].

The effective absorption of anthocyanins in the visible region of the spectrum reduces the damage caused by the HIL. Moreover, they function as antioxidants, scavenging ROS and thereby supporting cellular defense mechanisms against oxidative stress [31]. Low anthocyanin content might induce enhanced susceptibility to HIL and UV-induced photodamage of cell components, including nucleic acids, proteins, and lipids [32]. Balancing pigment content is therefore imperative for plant health, especially in fluctuating environmental light conditions. Therefore, while HIL can be a stressor for plants, it also induces complex mechanisms to protect plants from this stress. However, prolonged exposure to HIL can decrease plant productivity [33].

Earlier studies showed that tomato high-pigment (*hp*) mutants are capable of increased production of flavonoids and photosynthetic pigments in fruits. Additionally, it has been shown that manipulation of component genes of light signal transduction, such as *HY5* and *COP1,* may lead to changes in the accumulation of carotenoids in fruits [34]. COP1 is an E3 ubiquitin ligase that targets photomorphogenesis enhancers for degradation. In the absence of light, COP1 is active, targeting proteins such as HY5. Upon light exposure, its activity is inhibited, prompting photomorphogenesis [35]. Changes in the DET1 protein content were a negative factor in photomorphogenesis, which led to the formation of fruits with a high carotenoid content [36]. DET1 is a protein that is a negative regulator of photomorphogenesis. Similar to COP1, DET1 functions in the dark to prevent premature photomorphogenesis. DET1 works alongside COP1 in ubiquitinating and degrading light-signaling components [37]. DDB1 (damaged DNA binding protein 1) is a component of light signal transduction machinery involved in the repression of photomorphogenesis in darkness by participating in the CDD complex, a complex probably required to regulate the activity of ubiquitin-conjugating enzymes (E2s). The latter plays a role in the repression of photomorphogenesis and is probably mediated by ubiquitination and subsequent degradation of photomorphogenesis-promoting factors such as HY5, HYH, and LAF1 [38].

Influencing the pigmentation of ripe fruits and influencing the light-signaling pathway may be an effective means of improving the nutritional value of fruits [39]. Conversely, the *lp* 3617 mutant, characterized by its low pigment content, is linked to the gene identified as an ABI3-interacting protein [40,41]. This protein functions as a transcription factor in the abscisic-acid-signaling pathway. Furthermore, studies have demonstrated that this transcription factor also plays a role in plastid biogenesis. Disruptions in its function can result in reduced pigment content [42]. However, it is unclear to what extent such mutants are adapted to photoinhibition caused by HIL.

It is important to note that specific responses may vary among plant species and under different environmental conditions [43]. In addition, although we have gained significant insight into the molecular mechanisms that regulate the biosynthesis of phenolic compounds, many aspects of this process are still not fully understood and are the subject of ongoing research. The synthesis and accumulation of phenolic compounds is highly regulated by plant genetic and metabolic networks and can be influenced by various external factors, including light intensity [44]. In connection with the foregoing, the aim of this work was to study how deficiency or excess of photosynthetic and other pigments in mutant tomato plants can affect their ability to adapt to HIL.

## 2. Materials and Methods

### 2.1. Plant Material and Experimental Design

In the experiments, wild-type (WT) *Solanum lycopersicum* L. plants (Moneymaker cultivar, LA2706) and a photomorphogenetic high-pigment (*hp*) 3005 mutant with an increased content of pigments, such as isoprenoids and phenolic compounds (mutation of the *De-etiolated 1* (*DET1*) gene), and a low-pigment (*lp*) 3617 mutant (mutation of the *Abscisic-acid-insensitive 3* (*ABI3*) gene) with a reduced pigment content were used. The seeds were obtained from the Tomato Genetics Resource Center (TGRC) (University of California, Davis, CA, USA). The plants were grown for 30 days in a thermostatically controlled chamber with a 12 h photoperiod at a temperature of 23 ± 1 °C during the day and night. Then, the plants were cloned using cuttings and grown for 15 days. Then, some of the 45 d plants were grown under white fluorescent lamps (WFL) (Philips, Pila, Poland) at a light intensity of 250 μmol photons m^−2^s^−1^ in 8 cm × 8 cm × 10 cm vessels filled with perlite. Throughout the cultivation season, the plants were watered with half-strength Hoagland solution. Plants were continuously irradiated with high-intensity light for 72 h using broad-spectrum white LEDs (intensities of 1000 and 2000 ± 50 µmol (photons) m^−2^s^−1^) (450 nm + 580 nm) (Epistar, Hsinchu, Taiwan). The spectral characteristics of the light sources were determined using an AvaSpecULS2048CL-EVO spectrometer (Avantes, Apeldoorn, The Netherlands). At the end of the experiment, young leaves formed under appropriate light conditions were selected for analysis and fixed in liquid nitrogen. Leaves of original plants grown under fluorescent lamps were used as a control. All the remaining analyses, including microscopy, were conducted until and after 24 h and 72 h of HIL treatment. Six to ten of the most developed leaves from the second and third tiers were used for the analysis. Over the course of irradiation with light, the PSII activity was determined, and similar samples were taken for microscopic analyses and pigment determination.

### 2.2. Determination of Water Content, Photosynthetic and UV-Absorbing Pigments, Anthocyanins and Trolox Equivalent Antioxidant Capacity

The fresh weights were determined with an accuracy of 1 mg using an analytical balance (Scout Pro SPU123, Ohaus Corporation, Parsippany, NJ, USA). The dry weight was determined using an analytical balance (AB54-S, Mettler Toledo, Im Langacher Greifensee Switzerland) with an accuracy of 0.1 mg after drying the samples for 72 h at 70 °C to a constant weight. The water content was expressed as a percentage of fresh weight.

The chlorophyll *a* (Chl *a*), *b* (Chl *b*) and carotenoid (Car) contents were determined using the Lichtenthaler method [45]. The samples were triturated with 80% acetone in the dark. The absorbance of the solutions after centrifuging the samples was measured with a Genesys 10 UV–Vis spectrophotometer (Thermo Fisher Scientific, Waltham, MA, USA) at wavelengths of 470, 646, and 663 nm. The content of the photosynthetic pigments was determined using the Lichtenthaler formulas: Chl *a* = 12.25 × A663 − 2.79 × A646; Chl *b* = 21.50 × A646 − 5.10 × A663; Car = (1000 × A470 − 1.82 × Chl *a* − 85.02 × Chl *b*)/198.

UV-absorbing pigments (UAPs, mainly flavonoids) were isolated from fresh leaves by a previously described method [46]. Leaf cuts were kept for 24 h in acidic methanol (ethanol:water:HCl, 78:20:2) at 4 °C. Then, the optical density was determined at 327 nm and recalculated per 1 g (FW).

Anthocyanins were extracted and determined spectrophotometrically [47]. Briefly, 0.10–0.15 g of leaf mass per sample was ground in liquid nitrogen and incubated in 600 μL of extraction buffer (methanol containing 1% HCl) in an ultrasonic bath for 15 min and overnight at 4 °C in the dark.

The antioxidants of low molecular weight were isolated using 80% methanol from leaves that were pulverized in liquid nitrogen. Their antioxidant capacity, represented as Trolox equivalent antioxidant capacity (TEAC), was evaluated spectrophotometrically, based on the procedure established by Re et al., 1999 [48]. This method involves the interaction of methanolic extracts with 2,2′-azino-bis [3-ethylbenzothiazoline-6-sulfonic acid] diammonium salt (ABTS) (procured from Sigma-Aldrich, Burlington, MA, USA, with the CAS number 30931-67-0). The antioxidant potency of these low molecular weight compounds was denoted in µmol Trolox/g (FM).

### 2.3. HPLC Analysis of Carotenoids

The analysis of carotenoids was conducted following previously established methods [49,50]. Utilizing an HPLC apparatus (Shimadzu, Kyoto, Japan), which included (1) an LC-10ADVP pump complemented by an FCV-10ALVP module, (2) an SPD-M20A diode matrix detector, and (3) a CTO-20 AC thermostat. Carotenoid separation was achieved using a 4.6 mm × 250 mm reversed-phase column (Agilent Zorbax SB-C18, Agilent, Santa Clara, CA, USA), maintained at 22 °C. Carotenoids were identified based on their retention times and absorption spectra. Quantification was executed by comparing the peak area of each carotenoid in the 270–800 nm region against the cumulative peak areas of all carotenoids, taken as 100%. Calculations were facilitated using the LC-solution program (Shimadzu, Kyoto, Japan), applying the molar extinction coefficients as described in previous literature [51].

### 2.4. Measurements of CO_2_ Gas Exchange and Photochemical Activity

The photosynthetic rate (Pn, μmol CO_2_ m^−2^s^−1^) was determined in a closed system under light conditions using an LCPro + portable infrared gas analyser from ADC BioScientific Ltd. (Hoddesdon, UK) that was connected to a leaf chamber with an area of 6.25 cm^2^. The rate of photosynthesis of the leaves was determined at a saturating light intensity of 1000 μmol (photons) m^−2^s^−1^.

The fluorescence induction curves were measured with a mini-PAM II fluorometer (Walz, Effeltrich, Germany) on plants adapted to the dark (30 min). Blue LEDs (maximum 474 nm) were used to provide the measuring light (0.5 μmol (photons) m^−2^s^−1^), actinic light (100 μmol (photons) m^−2^s^−1^) and saturating pulses (maximum 474 nm, 3000 µmol (photons) m^−2^s^−1^ and 800 ms duration). The parameter calculations on the basis of Chl fluorescence data were performed using WinControl-3 v3.32 software (Walz, Effeltrich, Germany). The values for F_0_, F_v_, F_m_, F_m_′, and F_0_′, as well as the PSII maximum (F_v_/F_m_) and effective Y(II) (F_m_′ − F_t_)/F_m_′ photochemical quantum yields and nonphotochemical quenching (NPQ) (F_m_ − F_m_′)/F_m_′ were determined [52]. Here, F_m_ and F_m_′ are the maximum Chl fluorescence levels under dark- and light-adapted conditions, respectively. F_v_ is the photoinduced change in fluorescence, Ft is the steady-state level of Chl fluorescence, and F_0_ is the initial Chl fluorescence level. Additionally, Y(NO) and Y(NPQ) are quantum yields of non-regulated and regulated non-photochemical energy dissipation in PSII, respectively, and were determined [52].

### 2.5. RNA Extraction and Quantitative RT-PCR

RNA isolation was performed according to the TRIZOL^®^ (Sigma, St. Louis, MO, USA) method. cDNA synthesis and qRT-PCR analysis of gene expression patterns were performed according to the manufacturer’s protocol. The list of gene-specific primers is given in Appendix A according to https://phytozome-next.jgi.doe.gov/database the access date 1 February 2023. The transcript levels were normalized to the expression of the Tubulin alpha chain (Tubulin 1 A0A3Q7F8W6). Gene expression patterns were assessed using the CFX96 Touch™ real-time PCR detection system (Bio-Rad, Hercules, CA, USA). Gene-specific primers (Appendix A) for DNA-binding protein 1 (*DDB1*, NM_001247346.1); anthocyanidin synthase (*ANS*, NM_001374394.1); phytoene synthase (*PSY*, NM_001247883.2); chalcone synthase (*CHS*, NM_001247104.2); transcription factor elongated hypocotyl 5 (*HY5*, NM_001247891.2), transcription factor phytochrome-interacting factor 4 (*PIF4*, NM_001308008.1), E3 ubiquitin-protein ligase (*COP1*, NM_001247118.2), phenylalanine ammonia-lyase 1 (*PAL1*, XM_004249510.4), flavonol synthase (*FLS*, XM_004242121.4), and De-etiolated1 (*DET1*, NM_001247219.2) were selected using nucleotide sequences from the National Center for Biotechnology Information (NCBI) database (www.ncbi.nlm.nih.gov, USA accessed on 2 January 2023), https://www.uniprot.org/ (accessed on 2 January 2023), and https://phytozome-next.jgi.doe.gov/ (accessed on 2 January 2023), with Vector NTI Version 9 software (Invitrogen, Waltham, MA, USA). Transcript levels were normalized according to *Tubulin 1* gene expression. Gene expression in WT was assigned a value of 1.

### 2.6. Light Microscopyver

Thin cross-sectional slices of leaves were carefully prepared using a secure razor blade. These slices were then placed on a glass slide, gently immersed in a few droplets of demineralized water, and subsequently covered with a cover glass for examination. Observations were carried out using an Olympus CX41 microscope (Olympus, Tokyo, Japan). For documentation, microphotographs of the sections were captured utilizing an MS60 color video camera (MSHOT, Guangzhou, China).

### 2.7. Statistical Analysis

Fluorescence, photosynthetic, and transpiration rate measurements were conducted on four to six biological replicates from fully developed leaves of the middle tiers. Each plant, preserved in liquid nitrogen, was considered a separate biological replicate. A total of three biological replicates were utilized for determining pigment contents—including anthocyanins and UAPs—as well as for evaluating the TEAC value and conducting gene expression analysis. Each of these experiments comprised at least three parallel independent measurements to ensure reliability. Statistical significance of differences between groups was assessed using one-way analysis of variance (ANOVA), followed by post-hoc analysis employing Duncan’s method, facilitated by SigmaPlot 12.3 (Systat Software, San Jose, CA, USA). Statistical significance is denoted by letters, indicating meaningful differences between variants at a *p*-value of less than 0.05, unless specified otherwise. The presented data encompass arithmetic means complemented by standard errors for a comprehensive understanding of the variability and reliability of the findings.

## 3. Results

### 3.1. Leaf Morphology, Water Content, Photosynthetic Pigments, UV-Absorbing Pigments, Anthocyanins and Trolox Equivalent Antioxidant Capacity

HIL significantly affected the leaf structure of the studied mutant lines. The 3005 mutant was much more capable of accumulating anthocyanins than the 3617 mutant. The 3617 mutant produced smaller leaves with a curved leaf blade (Figure 1).

In the course of the studies, the antioxidant activity of leaf extracts was studied. High-intensity light at 1000 µmol (photons) m^−^^2^s^−^^1^ increased TEAC capacity approximately 3 times in all variants, while the maximum value of activity was in the *hp* mutant (Table 1).

After 72 h, this trend in TEAC values continued. At an intensity of 2000 µmol (photons) m^−2^s^−1^ activity of non-enzymatic antioxidants in the *hp* mutant 3005 increased and exceeded the starting point of WT by almost 6 times and the activity of WT by 3.4 times (Table 1). At the same time, in other mutants and WT, by 72 h, a decrease in the values of TEAC extracts was observed compared with the corresponding indicators at lower doses of HIL (Table 1).

At the starting point, the content of chlorophyll and carotenoids in the *hp* mutant was more than twice as high as that in *lp* and 1.3–1.6 times higher than that in WT (Table 1). At 1000 µmol (photons) m^−2^s^−1^ after 24 h, an increase in pigment content was observed in *lp* and WT, while in *hp*, the chlorophyll and carotenoids content decreased (Table 1). After 72 h at a light intensity of 1000 µmol (photons) m^−2^s^−1^, the levels of all pigments decreased in the mutants and WT. At the 24 h point at 2000 µmol (photons) m^−2^s^−1^, the pigment content did not change in WT, and in *hp,* a decrease of 30% was observed relative to the starting point. In *lp,* on the contrary, a slight increase was observed, and after 72 h in 3617 and WT, a decrease in the content of photosynthetic pigments was observed. However, in the *hp* 3005 mutant, an increase in content was recorded, and the maximum pigment content among the studied options was 3022 μg g^−1^ (Table 1). It is worth noting that in the WT, the ability to accumulate anthocyanins was lower than that in the *hp* mutant. It is also worth noting that in mutant 3005, the thickness of the leaf blade increased, and by the end of the experiment, it was approximately 2 times higher than this parameter in 3617 and WT (Table 1). At the same time, in mutant 3005, the percentage of water in the leaves was highest among the studied tomato lines (2000 µmol (photons) m^−2^s^−1^ after 72 h), and in mutant 3617, in contrast, the percentage of dry mass increased (Table 1).

### 3.2. Qualitative and Quantitative Composition of Carotenoids

Initially, the percentage composition of all carotenoids was approximately similar. At a light intensity of 1000 µmol (photons) m^−2^s^−1^ in 3005, the percentage of neolutein increased, in 3617 lutein, and in WT zeaxanthin relative to the starting point (Figure 2).

Under conditions of 2000 µmol (photons) m^−2^s^−1,^ the main distinguishing feature of the 3617 mutant was the increase in neoxanthin in the first 24 h, as well as the maximum percentage of lutein after 72 h of irradiation (Figure 2). The remaining carotenoids showed a similar trend in abundance between mutants (Figure 2).

### 3.3. Measurements of CO_2_ Gas Exchange of Photochemical Activity

At the starting point of the experiment, the net photosynthetic rate (Pn) averaged 6–8 µmol CO_2_ m^−2^s^−1^ (Table 2). After 24 h of irradiation, the Pn value increased in all variants, and after 72 h of irradiation, it decreased compared to the 24 h time point (Table 2). At a light intensity of 1000 µmol (photons) m^−2^s^−1^ after 24 h of irradiation, the Pn value was at its maximum in mutant 3005 (21.8 µmol CO_2_ m^−2^s^−1^) (Table 2). After 72 h at 1000 µmol (photons) m^−2^s^−1^, the rate of photosynthesis decreased in the mutants and WT but reached a maximum in the 3005 mutant (Table 2). A similar trend was observed when plants were irradiated with 2000 µmol (photons) m^−2^s^−1^. After 72 h of irradiation at 2000 µmol (photons) m^−2^s^−1^, maximum photosynthesis (10.9 µmol CO_2_ m^−2^s^−1^) was observed in the 3005 mutant, while in the WT and 3617 mutant, the Pn value was within 5–7 µmol CO_2_ m^−2^s^−1^. Apparently, mutant 3005 is the most resistant to the action of HIL, since its effect of photoinhibition of photosynthesis was the least (Table 2).

In addition to the Pn index, the values of the maximum quantum yield of PSII (F_v_/F_m_) were approximately the same at the starting point, and after irradiation with HIL, they decreased in 3617 at 1000 and 72 h and at 2000 µmol (photons) m^−2^s^−1^ at 24 h and 72 h, as well as at 2000 µmol (photons) m^−2^s^−1^ and 72 h for WT. At the same time, the F_v_/F_m_ indicators of 3005 in all variants remained approximately at the original level. The performance of PSII (PI_ABC_) and the effective quantum photochemical yield of PSII Y(II) differed little from each other at the starting point (Table 2).

However, during the experiment, the parameters characterizing photosynthetic activity decreased significantly in all the studied plants but to a lesser extent in the 3005 mutant, in which these values were the highest among the variants at 72 h and 2000 µmol (photons) m^−2^s^−1^ (Table 2). This indicates that the degree of photoinhibition was the lowest in the 3005 mutant. Similar to other photosynthetic indicators, NPQ and DI_0_/RC, which characterize the dissipation of absorbed energy into heat, differed little at the starting point but increased noticeably after 24 h of irradiation_,_ both at 1000 and at 2000 µmol (photons) m^−2^s^−1^ (Table 2). However, no significant difference was found between mutants at 2000 µmol quanta and 72 h. On the other hand, there was a difference between 3005 and 3617 mutants in the Y(NO) value, which was higher in the 3617 mutant, but there was no difference in the Y(NPQ) value.

Thus, parameters reflecting PSII activity, such as PI_ABC_, Y(II), and F_v_/F_m_, decreased at maximum photoinhibition (2000 µmol (photons) m^−2^s^−1^ 72 h) to a lesser extent in the 3005 mutant. Additionally, under these conditions, the 3005 mutant showed a tendency toward the smallest increase in NPQ and DI_0_/RC values.

### 3.4. Transcript Levels of the Studied Genes

Initially, the transcript levels of the *PIF4, COP1,* and *PAL1* genes in *hp* mutant 3005 were higher than those in the control and *lp* mutant 3617. After 24 h at 2000 µmol (photons) m^−2^s^−1,^ the main difference between 3005 and WT was the increase in *HY5* expression by more than 14-fold (Figure 3). In the *lp* mutant, *PIF4* expression increased relative to that in the WT. After 72 h at 2000 µmol (photons) m^−2^s^−1^ in the mutants and WT, the expression of the *CHS*, *FLS*, and *HY5* genes increased 3–6 times relative to the WT initial point (Figure 3). A distinctive feature of the *hp* mutant was an increase in *ANS* transcription by 10 times relative to the WT initial point (Figure 3), which was consistent with the high anthocyanin content.

### 3.5. The Structure of Leaf Tissues

Before the treatment, in *S. lycopersicum* mutant 3005 and WT 2706, anthocyanins were detected in some cells of the subepidermal layer of chlorenchyma in the leaf petioles and rachises (Figure 4A,B) and occasionally in some cells of the subepidermal parenchyma and collenchyma below the veins (Figure 4E) and subepidermal cells above the veins. In mutant line 3005, the accumulation of anthocyanins in the subepidermal cells was the most prominent, and they were also occasionally found in the basal cells of small non-glandular trichomes, in the cells at the base of large multicellular trichomes (type I glandular trichomes) and in some parenchyma cells surrounding the xylem vessels in leaf petioles. Anthocyanins were not visually detected in the leaves of the 3617 line (Figure 4C,F).

After 72 h of exposure, the accumulation of anthocyanins in the cells of the subepidermal parenchyma and collenchyma below the veins of the leaflets (Figure 4J,K) and in the cells of the subepidermal layer of chlorenchyma in the leaf petioles and rachises (Figure 4G,H) increased in plants of all lines except 3617 (Figure 4I,L) but most prominently in the 3005 line. In the 3005 and 2706 lines, the epidermal cells in these regions sometimes showed light purple–violet coloring, and anthocyanins also accumulated in the subepidermal cells at the abaxial side of the leaflets close to the veins (Figure 4O,P), in the cells at the base of large multicellular trichomes (type I glandular trichomes) (Figure 4N,Q), in the basal cells of small non-glandular trichomes (Figure 4M,O,P) and in subepidermal cells above the veins. Moreover, in 3005, the accumulation of anthocyanins was noted in the cells of the palisade mesophyll (Figure 4V) and occasionally in the subepidermal layer of spongy mesophyll in the regions between the veins of the leaflets. In the 3005 line, anthocyanins were also detected in some parenchyma cells close to the vascular tissues in leaf petioles and rachises.

## 4. Discussion

High-intensity light conditions can pose significant challenges for plants, necessitating a range of adaptive responses to mitigate potential damage. The adaptive mechanisms in plants with varying pigment content, specifically low-pigment and high-pigment plants, differ in several ways of adaptation, one of which is the absorption of light and its distribution in leaf tiers. Plants with high pigment content tend to absorb more light, converting it into energy for photosynthesis. This can be advantageous under low-light or moderate-light conditions, as these plants can maximize the use of available light [53]. However, under HIL, increased absorption can lead to photoinhibition or damage to the photosynthetic machinery [3]. On the other hand, low-pigment plants absorb less light, potentially reducing the risk of photodamage under intense light conditions, but in this case, photosynthesis decreases. Photoprotection mechanisms also vary among plants with different levels of pigments. High-pigment plants often possess a broader range of photoprotective mechanism compounds, such as carotenoids and flavonoids, which can serve as optical filters and manifest antioxidant activity preventing potential damage to the PA [54]. Previously, we confirmed this in another tomato photomorphogenetic mutant [53]. In our experiment, in the studied tomato lines, the main differences occurred after three days of maximum HIL irradiation (Table 1 and Table 2). Thus, the effective quantum yield of PSII Y(II), PSII performance index PI_ABS_, and net photosynthetic rate increased in the *hp* mutant at 2000 µmol (photons) m^−2^s^−1^ 72 h (Table 2). This indicates the onset of PA adaptation to photoinhibition caused by HIL. In response to HIL, plants may adjust the efficiency of their photosynthetic processes. High-pigment plants might modulate their photosystem activities and the size of PSII light-harvesting antennae to prevent overexcitation and consequent damage. Thus, low-pigment plants might rely more on nonphotochemical quenching (NPQ), a process by which excess energy is dissipated into heat and has a reduced net photosynthetic rate to prevent ROS formation [55] and may reduce their photosynthetic efficiency under intense light due to their reduced light absorption [53,56].

The data obtained in our work confirm the presence of different adaptation strategies to photoinhibition in *hp* and *lp* mutants (Table 1 and Table 2). Light energy excites photosynthetic reaction centers and is absorbed by light-harvesting complexes [57]. The generation of ROS and radicals during photocatalytic processes in PSII is dangerous for photosystem pigment–protein complexes. In damaged PSII reaction centers, charge separation is disrupted, and excited chlorophylls transfer energy for pathways other than photosynthetic pathways to prevent the formation of singlet oxygen [58]. According to our results, the *lp* mutant increased fluorescence quenching (NPQ and Y(NO)) (Table 2), which was observed to a lesser extent in the *hp* mutant (Table 2). Energy can be redirected to functional reaction centers through the LHCII system or dissipated into heat when PSII is inhibited [58]. The antenna size effectiveness is adaptively modified based on light intensity and the plant metabolism capacity to utilize the gathered energy. Surplus energy that is not consumed is dispersed thermally through NPQ processes [11], diminishing the light-harvesting efficiency when the incoming light energy surpasses what the plant can metabolically utilize. The LHCII system’s dynamic nature likely holds significant influence in managing the excitation energy transfer during adaptation to photoinhibition [47].

PSII exhibits sensitivity to an overflow of light energy. Within the LHCII system, operational molecular mechanisms result in either the induction of photodamage by PSII or a defense against such damage through the excess energy dissipation by NPQ. This points towards the presence of a violaxanthin–zeaxanthin-reliant safeguard, protecting PSII from photoinhibition by facilitating the quenching of triplet chlorophylls in the PSII-LHCII complex [59]. In our studies, the main difference observed in *hp* mutants was the increase in β-carotene content at 72 h and 2000 µmol (photons) m^−2^s^−1^ (Figure 2). β-Carotene is a crucial carotenoid present in chloroplasts, particularly within photosystems [60]. Its functions include assisting in light absorption and protecting chlorophyll molecules from photooxidation by quenching singlet oxygen species. The rise in β-carotene concentration with increased light intensity is a part of the plant’s adaptive response to shield the photosynthetic apparatus from potential damage caused by ROS under high light conditions [61]. Additionally, carotenoids, including violaxanthin and β-carotene, play a role in maintaining the structural integrity of photosynthetic complexes [62]. Increased accumulation of these pigments might be a response to ensure resistance of the photosynthetic apparatus under high light stress [61].

The synthesis and interconversion of carotenoids are regulated by complex feedback mechanisms based on the plant’s internal and external environment. HIL might induce enzymatic activity or transcriptional changes leading to enhanced production or stabilization of specific carotenoids [63]. In our work, we observed an increase in the expression of *CHS*, *FLS*, *ANS*, and *HY5* in both mutants, but the level of transcription of these genes in the *hp* mutant was higher than that in the *lp* mutant (Figure 3c). Another advantage of the *hp* mutant is the earlier expression of *HY5* (Figure 3b). The possible key role of *HY5* in hp mutants was considered in our previous studies [63].

Additionally, the difference in adaptation strategies to photoinhibition between the mutants studied was confirmed by different amounts of water and dry mass. Thus, the *hp* mutant formed thicker leaves (Table 1, Figure 4V), but this was due to the accumulation of water; in addition, in the *lp* mutant, the percentage of dry mass increased (Table 1). Plants may increase leaf thickness and reduce leaf area in response to HIL to reduce water loss. The resulting smaller, thicker leaves tend to hold less water. At high light intensities, the rapid production of assimilates (products of photosynthesis) can sometimes exceed the plant’s ability to use them for growth, resulting in reduced cell expansion and, as a result, reduced water uptake by cells, which was indicated in the case of the 3617 mutant.

Thus, high-intensity light triggers physiological reactions that can lead to changes in leaf mass. At the same time, mechanisms of increased transpiration lead to a decrease in water content in cells. As a result, we can say that increasing the percentage of water in cells in the *hp* mutant is ultimately a more successful mechanism because it allows the maintenance of the net photosynthetic rate and growth under light stress conditions. This was also confirmed with our data (Table 2).

Stress-induced antioxidant production is another key mechanism that can protect plants from photoinhibition under HIL conditions that can lead to the formation of ROS, damaging cellular structures [64]. Plants respond by producing antioxidants to neutralize these ROS. While both low-pigment and high-pigment plants produce antioxidants, high-pigment plants might produce them in larger amounts due to their potential increased ability to neutralize light-generated ROS [9]. Our study showed that the 3005 mutant exhibited a higher TEAC value than the *lp* mutant (Table 1). These findings suggest a correlation between changes in photosynthetic activity—both at the level of photochemical processes and CO_2_ gas exchange—and variations in the activity of low molecular weight antioxidants. Thus, the 3005 mutant, when exposed to HIL at its high dose, demonstrated the smallest photoinhibition, which, as we suggest, is associated with high pigment content and a thicker leaf (Table 2), which also agrees with previously obtained data [63].

It is also known that pigments in leaves can function not only as potent antioxidants but also as light optical filters, modulating light flow to photosystems [16,18]. In particular, high contents of anthocyanin play an important role in protection against photodamage by visible radiation [16,65]. Additionally, some carotenoids can play an important role in screening light in the blue–green region of the spectrum [63,66]. We observed high values of TEAC and elevated contents of anthocyanins and carotenoids in the 3005 mutant (Table 1, Figure 2). In particular, a significant difference in pigments between 3005 and 3617 mutants was indicated at an intensity of 2000 µmol (photons) m^−2^s^−1^. For example, at 72 h, the HIL anthocyanin content was higher in the 3005 mutant than in the 3617 mutant by almost 40 times and carotenoids by 1.5 times. A much higher content of anthocyanins under the above conditions was also observed in the parenchyma and epidermis of the 3005 mutant compared to the 3617 mutant. Elevated expression of light-regulated genes, such as *CHS*, *ANS*, and *FLS*, also suggests an increase in flavonoid contents (Figure 3). It can be assumed that a high content of all these pigments contributes to the resistance of PA to long-term HIL.

The contribution to elevated contents of the above pigments under high irradiance is due to enhanced expression of genes linked to enzymes of pigment biosynthesis. We indicated a decline in *COP1* and *DDB1* transcript levels, leading to deficiency of the CDD complex (COP1, DDB1, DET1), which plays a role in the proteasomal degradation of certain photomorphogenesis transcription factors (e.g., HY5, HYH, LAF1, HFR1). This complex affects pigment formation [63]. Therefore, a possible reduction in the complex content may boost the transcription of the key transcription factor HY5, which can bind to promoters of genes encoding enzymes of pigment biosynthesis such as *POR* and *CHS*, enhancing their transcription and accelerating the synthesis of essential pigments. In addition, *HY5* can induce the expression of many other light-regulated genes [67], which can also contribute to the mechanisms of enhanced resistance of the 3005 mutant under high irradiance.

## 5. Conclusions

The adaptation of plants to HIL is multifaceted, with the responses being finely tuned based on their inherent pigment contents. Thus, under moderate stress caused by HIL, photoinhibition was not significantly different in *lp* and *hp* mutants; that is, the protective role of pigments is not clearly expressed. However, under severe stress caused by long-term exposure to HIL, less photoinhibition was observed in the mutant with an increased content of pigments. The increase in violaxanthin and beta-carotene levels under high light intensity reflects a plant multipronged strategy ensuring photoprotection and maintaining the structural and functional integrity of the photosynthetic apparatus. The specifics of these changes depend on the duration of light exposure, the species or variety of the plant, and other environmental conditions during the experiment. The *hp* mutation led not only to an increase in the content of carotenoids important for functioning and PSII protection but also to an increase in TEAC and the content of the various pigments, which was ultimately expressed in an increase in the intensity of photosynthetic activity and a decrease in photoinhibition. We also assume that the observed phenomena are based on earlier expression of *HY5*, which provides the necessary signaling and affects many of the observed processes. We hypothesize that not only the high content of anthocyanins and carotenoids but also the increased leaf thickness (pleiotropic effect) found in the 3005 mutant with prolonged exposure to HIL play a major role in the protection of the photosynthetic apparatus from HIL. The dynamic interplay of biochemical, physiological, and morphological processes demonstrates the adaptation and flexibility of plants in changing environmental conditions.

## Figures and Tables

**Figure 1 cells-12-02569-f001:**
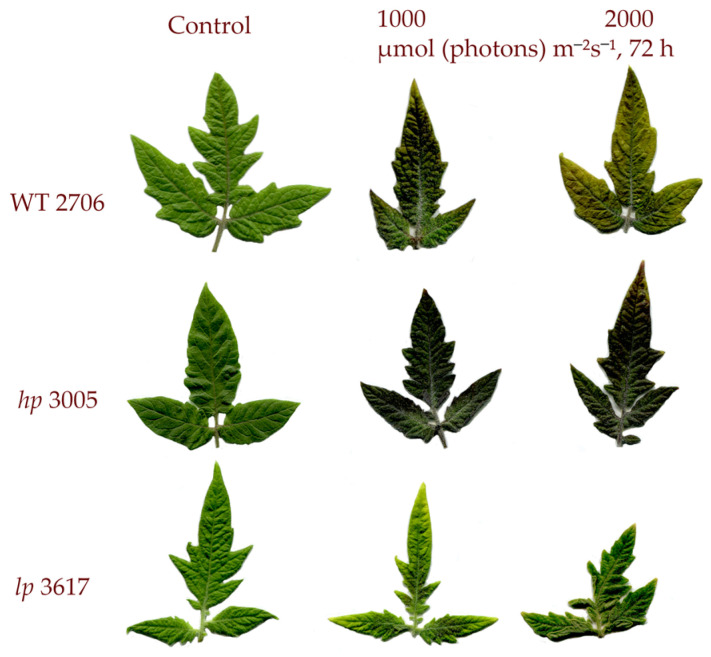
Appearance of leaves of tomato *hp*, *lp* mutants and WT before and after HIL for 72 h at 1000 and 2000 µmol (photons) m^−^^2^s^−^^1^ irradiation.

**Figure 2 cells-12-02569-f002:**
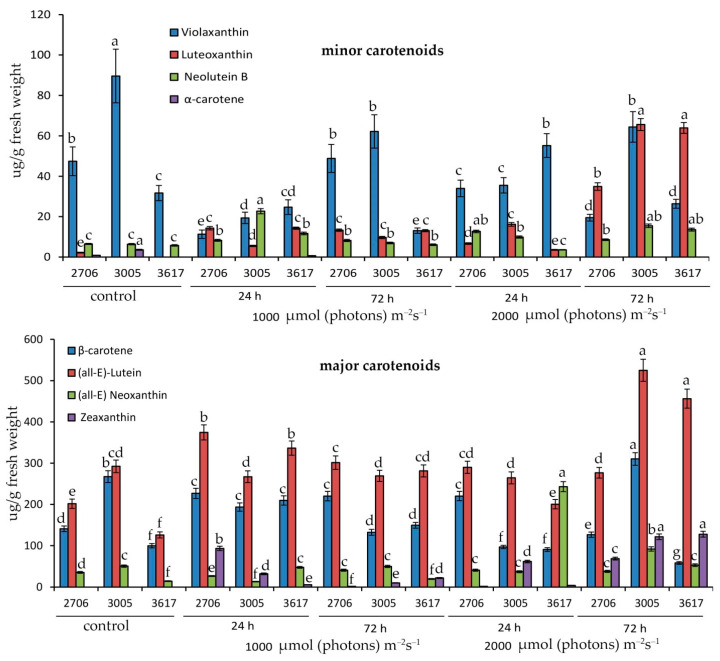
Effect of high-intensity light at 1000 and 2000 µmol (photons) m^−2^s^−1^ for 24 and 72 h on the content of basic carotenoids in the leaves of *S. lycopersicum hp* 3005, *lp* 3617 mutants and 2706 WT. Different letters designate statistically significant differences in mean values at *p* < 0.05 (ANOVA followed by Duncan’s method) among certain carotenoids throughout the experiment. Mean values ± standard errors, *n* = 3.

**Figure 3 cells-12-02569-f003:**
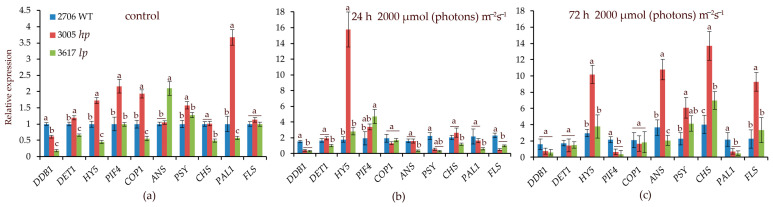
Transcript levels of the main photoreceptor-regulated genes involved in light signaling (*DDB1*, *HY5*, *PIF4*, *COP1*, *DET1*) and in the biosynthesis of secondary metabolites such as flavonoids and carotenoids (*CHS*, *PAL*, *ANS*, *PSY*, *FLS*) under high-intensity light of 2000 µmol (photons) m^−2^s^−1^ under control conditions (**a**), as well as after 24 h (**b**) and 72 h (**c**) of HIL irradiation. Transcript levels were normalized to *Tubulin1* gene expression. Gene expression in WT in control conditions was taken as 1 unit. Different letters designate statistically significant differences in mean values at *p* < 0.05 (ANOVA followed by Duncan’s method) among the studied tomato lines within a specific gene. Mean values ± standard errors, *n* = 3.

**Figure 4 cells-12-02569-f004:**
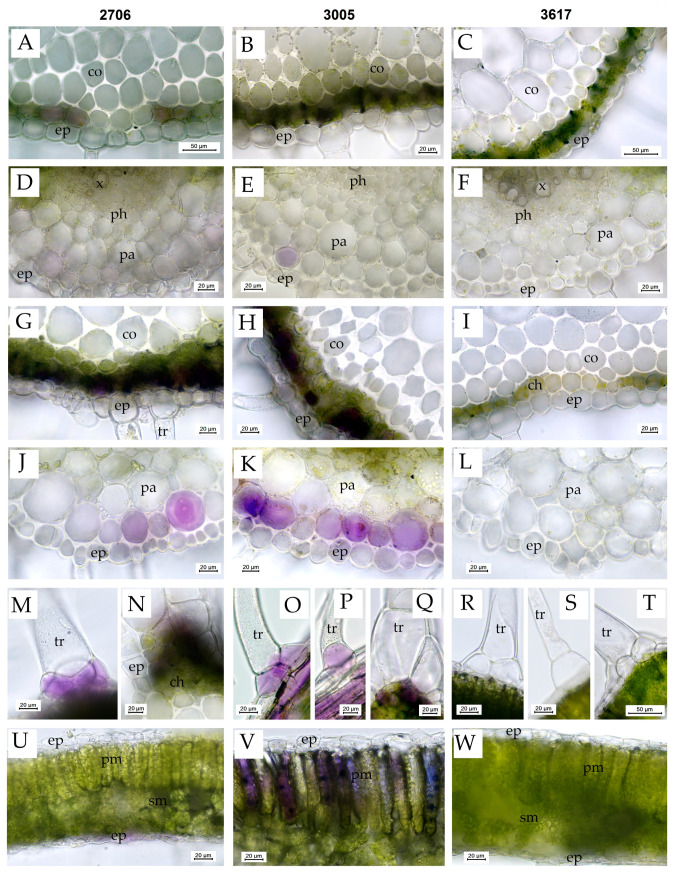
Vital sections of *S. lycopersicum* leaves showing the distribution of anthocyanins as well as the structure of various tissues before and after 72 h of HIL treatment. (**A**–**F**)—plants before HIL treatment; (**G**–**W**)—plants after 72 h of HIL treatment 2000 µmol (photons) m^−^^2^s^−^^1^. (**A**,**D**,**G**,**J**,**M**,**N**,**U**)—WT 2706; (**B**,**E**,**H**,**K**,**O**,**P**,**Q**,**V**)—*hp* mutant 3005; (**C**,**F**,**I**,**L**,**R**,**S**,**T**,**W**)—*lp* mutant 3617; (**A**–**C**,**G**–**I**)—cross sections of leaf petioles (note the increase in anthocyanin accumulation in the subepidermal layer in HIL-treated lines 2706 and 3005); (**D**–**F**,**J**–**L**)—leaf blade cross sections in the vein region below the vein (note the increase in anthocyanin accumulation in the subepidermal layer in HIL-treated lines 2706 and 3005); (**M**,**O**,**P**)—the accumulation of anthocyanins in the basal cells of the trichomes in HIL-treated lines 2706 and 3005; (**N**,**Q**)—the accumulation of anthocyanins in the cells at the base of large glandular trichomes, type I; (**R**,**S**,**T**)—no accumulation of anthocyanins in the basal cells of the trichomes in HIL-treated *lp* mutant 3617; (**U**–**W**)—leaf blade cross sections (note the accumulation of anthocyanins in the palisade mesophyll cells in HIL-treated 3005 *hp* mutant. Designations: ch—chlorenchyma, co—collenchyma, ep—epidermis, pa—parenchyma, ph—phloem, pm—palisade mesophyll, sm—spongy mesophyll, tr—trichome, vb—vascular bundle, x—xylem.

**Table 1 cells-12-02569-t001:** Effect of high-intensity light (1000 and 2000 µmol (photons) m^−2^s^−1^ for 72 h) on the content of the main photosynthetic pigments: chlorophyll *a* (Chl *a*), chlorophyll *b* (Chl *b*), total carotenoids (Car) (µg g^−1^ FW), the Trolox equivalent antioxidant capacity (TEAC) of total low molecular weight antioxidants (µmol Trolox g^−1^ FW), and on the leaf thickness (µm) of *hp* 3005, *lp* 3617 mutants and WT 2706. Different letters designate statistically significant differences in mean values at *p* < 0.05 (ANOVA followed by Duncan’s method) among the studied tomato lines within a specific experimental point. Mean values ± standard errors, *n* = 4. Within each column of the table, a color-coded scheme has been employed: red signifies an increase in the metric, blue denotes a decrease, and white indicates no change.

Variant	Light Intensity	Time, h	Chl *a*	Chl *b*	Chl *a + b*	Car	TEAC	Leaf Thickness, µm	Antocyanin µg g^−1^ FW	UAPs	H_2_O, %	DW, %
**2706**			1760 ± 70b	657 ± 26b	2417 ± 97b	435 ± 17b	25.5 ± 1.0a	108 ± 10a	0.45 ± 0.15b	4.4 ± 1.4c	84.4 ± 0.3b	15.6 ± 0.5b
**3005**	Control	**0**	2317 ± 93a	788 ± 32a	3105 ± 124a	711 ± 28a	20.3 ± 0.8b	110 ± 8a	1.33 ± 0.07a	6.7 ± 0.9b	87.8 ± 0.5a	12.2 ± 0.6c
**3617**			1120 ± 45c	380 ± 15c	1500 ± 60c	278 ± 11c	15.3 ± 0.6c	111 ± 12a	0.15 ± 0.02c	12.1 ± 2.1a	83.2 ± 0.8c	16.8 ± 0.3a
**2706**			2120 ± 85a	839 ± 34a	2959 ± 118a	753 ± 30a	53.1 ± 2.1b	115 ± 11a	1.88 ± 0.09b	7.1 ± 0.8b		
**3005**		**24**	1757 ± 70b	658 ± 26c	2415 ± 97b	553 ± 22b	66.5 ± 2.7a	121 ± 9a	2.41 ± 0.21a	4.3 ± 0.7c		
**3617**	1000 µmol (photons) m^−2^s^−1^		2258 ± 90a	784 ± 31b	3042 ± 122a	651 ± 26b	44.9 ± 1.8c	95 ± 11a	0.38 ± 0.03c	11.9 ± 1.8a		
**2706**			1683 ± 67a	535 ± 21a	2218 ± 89a	634 ± 25a	53.7 ± 2.2c	109 ± 8b	9.01 ± 0.52b	10.3 ± 0.5b	77.4 ± 0.7b	22.6 ± 0.7a
**3005**		**72**	1675.6 ± 67a	605 ± 24a	2280 ± 91a	541 ± 22b	113.9 ± 4.6a	130 ± 8a	12.24 ± 0.34a	5.2 ± 0.3c	79.9 ± 0.8a	20.1 ± 0.8b
**3617**			1524 ± 61a	654 ± 26a	2177 ± 87a	505 ± 20b	71.6 ± 2.9b	101 ± 10b	0.54 ± 0.14c	12.2 ± 0.3a	77.9 ± 0.4b	22.1 ± 0.4a
**2706**			1860 ± 74a	661 ± 26a	2521 ± 101a	607 ± 24a	48.5 ± 1.9b	119 ± 11a	2.9 ± 0.25b	8.2 ± 0.6b		
**3005**		**24**	1578 ± 63b	582 ± 23b	2160 ± 84b	522 ± 21b	56.0 ± 2.2a	123 ± 9a	5.7 ± 0.91a	5.8 ± 0.9c		
**3617**	2000 µmol (photons) m^−2^s^−1^		1996 ± 80a	641 ± 26a	2637 ± 106a	600 ± 22a	47.2 ± 1.9b	107 ± 13a	0.61 ± 0.05c	10.5 ± 0.7a		
**2706**			1428 ± 57b	579 ± 23ab	2008 ± 80b	573 ± 23c	43.8 ± 1.8b	97 ± 12b	11.01 ± 0.50b	13.4 ± 1.1b	78.8 ± 0.9b	21.1 ± 0.9b
**3005**		**72**	2405 ± 96a	617 ± 25a	3022 ± 121a	1194 ± 48a	149.7 ± 6.0a	197 ± 10a	18.22 ± 0.88a	6.1 ± 1.0c	81.1 ± 0.5a	18.8 ± 0.5c
**3617**			1458 ± 58b	724 ± 29a	2182 ± 87b	799 ± 32b	33.7 ± 1.4c	111 ± 12b	0.47 ± 0.02c	19.1 ± 1.6a	75.3 ± 0.8c	24.6 ± 0.8a

**Table 2 cells-12-02569-t002:** Effect of high-intensity light at 1000 and 2000 µmol (photons) m^−2^s^−1^ for 24 mutants and WT 2706: F_v_/F_m_—maximum quantum yield of PSII; Y(II) is the effective quantum yield of PSII; NPQ, nonphotochemical fluorescence quenching; DI_0_/RC, energy dissipation per 1 reaction center, PI_ABS_, PSII performance index; Pn (µmol CO_2_ m^−2^s^−1^), photosynthesis rate, Tr (transpiration rate, µmol H_2_O m^−2^s^−1^). Different letters designate statistically significant differences in mean values at *p* < 0.05 (ANOVA followed by Duncan’s method) among the studied tomato lines within a specific experimental point. Mean values ± standard errors, *n* = 3. Within each column of the table, a color-coded scheme has been employed: red signifies an increase in the metric, blue denotes a decrease, and white indicates no change.

Variant	Light Intensity	Time, h	NPQ	DI_0_/RC	PI_ABS_	Pn	Tr	F_v_/F_m_	Y(II)	Y(NO)	Y(NPQ)
**2706**			0.53 ± 0.06a	0.51 ± 0.02a	4.3 ± 0.3a	6.1 ± 0.6b	3.69 ± 0.16a	0.83 ± 0.01a	0.65 ± 0.02a	0.23 ± 0.01a	0.12 ± 0.02a
**3005**	Control	**0**	0.62 ± 0.06a	0.52 ± 0.03a	4.7 ± 0.3a	5.8 ± 0.7b	3.36 ± 0.17a	0.84 ± 0.01a	0.64 ± 0.02a	0.22 ± 0.01a	0.14 ± 0.01a
**3617**			0.53 ± 0.06a	0.48 ± 0.02b	5.0 ± 0.3a	7.9 ± 0.4a	2.81 ± 0.11b	0.83 ± 0.01a	0.60 ± 0.03a	0.27 ± 0.02a	0.14 ± 0.02a
**2706**			1.05 ± 0.11a	0.53 ± 0.02b	3.8 ± 0.4a	14.9 ± 0.8b	1.51 ± 0.13a	0.80 ± 0.03a	0.49 ± 0.06a	0.29 ± 0.02a	0.25 ± 0.03a
**3005**		**24**	0.81 ± 0.11a	0.73 ± 0.07a	2.8 ± 0.2ab	21.8 ± 2.2a	1.31 ± 0.15b	0.79 ± 0.02a	0.52 ± 0.03a	0.26 ± 0.01a	0.21 ± 0.03a
**3617**	1000 µmol (photons) m^−2^s^−1^		0.82 ± 0.12a	0.72 ± 0.03a	2.3 ± 0.3a	16.1 ± 0.7b	0.76 ± 0.08c	0.80 ± 0.02a	0.56 ± 0.04a	0.26 ± 0.02a	0.18 ± 0.03a
**2706**			0.67 ± 0.04b	0.63 ± 0.06c	1.9 ± 0.2a	7.5 ± 1.0 c	1.27 ± 0.16a	0.82 ± 0.03a	0.58 ± 0.03a	0.27 ± 0.04a	0.14 ± 0.01a
**3005**		**72**	0.76 ± 0.07b	0.72 ± 0.09b	2.5 ± 0.4a	13.8 ± 1.3a	0.5 ± 0.08b	0.79 ± 0.02a	0.55 ± 0.03a	0.26 ± 0.02a	0.19 ± 0.02a
**3617**			0.98 ± 0.05a	1.04 ± 0.09a	1.9 ± 0.3a	11.0 ± 0.2b	0.33 ± 0.06c	0.76 ± 0.02a	0.49 ± 0.03a	0.32 ± 0.01a	0.19 ± 0.03a
**2706**			1.19 ± 0.07a	0.75 ± 0.06b	1.7 ± 0.5a	15.7 ± 1.1a	0.95 ± 0.15a	0.77 ± 0.04a	0.47 ± 0.04a	0.28 ± 0.01a	0.22 ± 0.04a
**3005**		**24**	0.97 ± 0.02b	0.81 ± 0.12a	2.0 ± 0.2a	15.9 ± 1.0a	0.38 ± 0.05b	0.79 ± 0.03a	0.49 ± 0.04a	0.25 ± 0.02a	0.26 ± 0.02a
**3617**	2000 µmol (photons) m^−2^s^−1^		0.88 ± 0.07b	0.81 ± 0.05a	1.2 ± 0.3a	12.1 ± 0.3b	0.94 ± 0.11a	0.76 ± 0.03a	0.49 ± 0.03a	0.28 ± 0.02a	0.23 ± 0.03a
**2706**			1.12 ± 0.07a	1.07 ± 0.06a	0.9 ± 0.3b	4.7 ± 0.3c	0.56 ± 0.07a	0.65 ± 0.04b	0.43 ± 0.02b	0.33 ± 0.03ab	0.24 ± 0.03a
**3005**		**72**	0.87 ± 0.06b	0.76 ± 0.10b	2.2 ± 0.2a	10.9 ± 1.6a	0.19 ± 0.03c	0.78 ± 0.02a	0.53 ± 0.02a	0.27 ± 0.03b	0.20 ± 0.03a
**3617**			0.96 ± 0.06ab	0.96 ± 0.10ab	1.1 ± 0.2b	5.8 ± 0.2b	0.41 ± 0.06b	0.69 ± 0.03b	0.47 ± 0.02ab	0.37 ± 0.02a	0.17 ± 0.02a

## Data Availability

The datasets generated during and/or analysed during the current study are available from the corresponding author on reasonable request.

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
