# Peer review of "The Adaptive Role of Carotenoids and Anthocyanins in Solanum lycopersicum Pigment Mutants under High Irradiance"

_cells, 2023, doi:10.3390/cells12212569_

Round 1

Reviewer 1 Report

Comments and Suggestions for Authors

This is quite an extensive comparison of pigments, antioxidant activity, and also leaf structures of tomato mutants with different pigment-accumulating phenotypes under high-light treatment. Detailed results were collected in this study, and the explanation of the results is acceptable, also basically there is not too much novel information, especially since the DET1 tomato mutant has been well characterized. There are two major concerns that I believe the authors should work on before the manuscript can be considered for acceptance. 

First, data in the Tables are very difficult to read. The authors must try to make it clear which data were compared among each other for the differences. The current style of the tables is not readable.

Second, I doubt the light-dependent genes. First, these genes cannot be categorized as "light-dependent genes", because they are NOT. Second, how were these genes, from different metabolic and signaling pathways, picked?

Third, in Figure 3 (why is this figure no longer a color one?), if gene expression under different light intensities is one of the issues to be considered, the expression should not be normalized against the wild-type level in each of the panels.  Only the WT under the control condition can be set as 1, and all the others in all three panels should be compared with it. In this case, it will be possible to assess whether high light treatment positively or negatively regulates the expression of these genes.

Author Response

This is quite an extensive comparison of pigments, antioxidant activity, and also leaf structures of tomato mutants with different pigment-accumulating phenotypes under high-light treatment. Detailed results were collected in this study, and the explanation of the results is acceptable, also basically there is not too much novel information, especially since the DET1 tomato mutant has been well characterized. There are two major concerns that I believe the authors should work on before the manuscript can be considered for acceptance.

Answer: We sincerely thank the reviewer for their meticulous review and fair evaluation of our manuscript and are grateful for the time invested in reviewing our manuscript. However, we would like to respectfully disagree with the notion of limited novelty. The main studies were conducted under normal physiological conditions (Mustilli et al. 1999 https://doi.org/10.1105/tpc.11.2.145; Levin et al. 2006 https://doi.org/10.1560/IJPS_54_3_179). Our aim was to understand how high irradiance conditions affect photosynthetic processes and pigment content. In addition, our research included the impact of changes in pigment content and high irradiance on pro-/antioxidant balance and a number of physiological parameters together with the expression of some genes. Such a complex approach at high irradiance has a novelty. For a more exact illustration, we would like to note that to the best of our knowledge, the photomorphogenetic mutant 3005 is exclusively studied in our laboratory, and publications on this mutant are still in their nascent stages. While it has an orthologue, mutant 3006, classified as a det1 mutant, our experiments indicate significant differences between them, especially in their response to red and far-red light (will be published in 2024). Moreover, we believe that comprehensive studies involving mutant 3617 (mutation of the ABI3 gene)—encompassing its biochemical, molecular, and microscopic characterizations under light stress conditions—have yet to be conducted. If the Reviewer is aware of any existing work in this area, we would greatly appreciate the reference and will ensure it is duly cited in our manuscript.

  1. The data in the Tables are very difficult to read. The authors must try to make it clear which data were compared among each other for the differences. The current style of the tables is not readable.

Answer: We are grateful to the reviewer for the comment. The data compare three mutants at a single time point, as detailed in the table's caption. Recognizing that the table might be challenging to interpret, we have added color coding to each column. We hope this enhancement will facilitate a clearer understanding of the results and meet the reviewer's expectations.

  1. I doubt the light-dependent genes.

First, these genes cannot be categorized as "light-dependent genes", because they are NOT. How were these genes, from different metabolic and signaling pathways, picked?

Answer: We improved this point and wrote that these genes are photoreceptor-regulated genes. That is, we speak on photoregulation of plant gene expression. In our study, we initially sought to explore the genes implicated in network signaling, which, either directly or through clusters of transcription factors (which we also endeavoured to consider), possess the capability to influence the biosynthesis of both photosynthetic pigments and protective pigments, such as flavonoids and phenylpropanoids. We hypothesize that these elements may be involved in enhancing the tolerance of tomato plants with varying pigment contents to high-intensity light. We greatly appreciate the reviewer's constructive feedback. We trust that our response addresses the concerns raised and meets the reviewer's expectations.

We added this information to the Introduction section:

DDB1 (damaged DNA binding protein 1) is a component of light signal transduction machinery involved in the repression of photomorphogenesis in darkness by participating in the CDD complex, a complex probably required to regulate the activity of ubiquitin conjugating enzymes (E2s). The latter plays a role in the repression of photomorphogenesis and is probably mediated by ubiquitination and subsequent degradation of photomorphogenesis-promoting factors such as HY5, HYH and LAF1.

https://doi.org/10.1016/j.cbpa.2008.04.467

HY5 (elongated hypocotyl 5)

HY5 is a key transcription factor that promotes photomorphogenesis in light, acting downstream of the light receptor network although the transcription of light-induced genes. Specifically, it is involved in the blue light-specific pathway, suggesting that it participates in the transmission of cryptochrome (CRY1 and CRY2) signals to downstream responses.

https://doi.org/10.1016/j.molp.2017.03.012

PIF4 (phytochrome interacting factor 4)

PIF4 is a transcription factor that acts negatively in the phytochrome B signaling pathway. It may regulate the expression of a subset of genes involved in cell expansion by binding to the G-box motif.

https://doi.org/10.1016/j.cell.2015.12.018

COP1 (constitutively photomorphogenic 1)

COP1 is an E3 ubiquitin ligase that targets positive regulators of photomorphogenesis for degradation. In the dark, COP1 is active and targets proteins such as HY5 for degradation. In the light, its activity is suppressed, leading to photomorphogenesis.

https://doi.org/10.1104/pp.15.01184

DET1 (de-etiolated 1)

DET1 is a protein that is a negative regulator of photomorphogenesis.

Similar to COP1, DET1 functions in the dark to prevent premature photomorphogenesis. DET1 works alongside COP1 in ubiquitinating and degrading light signaling components.

https://doi.org/10.1046/j.1365-313X.1998.00078.x

CHS (chalcone synthase)

CHS is an enzyme that catalyzes the first committed step in flavonoid biosynthesis.

Light induces CHS expression, leading to the production of flavonoids, which have UV-absorbing properties and can protect plants from UV damage.

https://doi.org/10.1046/j.1469-8137.2001.00151.x

PAL (phenylalanine ammonia-lyase)

PAL is a key enzyme in the phenylpropanoid pathway, leading to the production of lignin, flavonoids, and other compounds. Light can induce PAL gene expression, contributing to increased synthesis of phenylpropanoid products.

https://doi.org/10.1016/j.postharvbio.2019.111069

ANS (anthocyanidin synthase)

ANS catalyzes the formation of anthocyanidins, which are precursors for anthocyanin pigments. Expression of the ANS gene can be upregulated by light, leading to increased anthocyanin production, which can provide protection against UV radiation and contribute to colouration.

https://doi.org/10.3390/ijms222011116

PSY (phytoene synthase)

PSY is a key enzyme in the carotenoid biosynthesis pathway. Light can induce PSY gene expression, leading to carotenoid production, which plays roles in photosynthesis and photoprotection.

https://doi.org/10.3390/ijms23116153

FLS (flavonol synthase)

FLS catalyzes the formation of flavonols from dihydroflavonols.

Light can stimulate FLS gene expression, leading to flavonol production, which can offer protection against UV and HIL damage.

https://doi.org/10.1007/s11103-004-6910-0

  1. In Figure 3 (why is this figure no longer a color one? ), if gene expression under different light intensities is one of the issues to be considered, the expression should not be normalized against the wild-type level in each of the panels. Only the WT under the control condition can be set as 1, and all the others in all three panels should be compared with it. In this case, it will be possible to assess whether high light treatment positively or negatively regulates the expression of these genes.

Answer: We are grateful to the reviewer for the comment. We have corrected Figure 3 according to the reviewer's recommendations.

Reviewer 2 Report

Comments and Suggestions for Authors

A. Ashikhmin and colleagues present an article on the correlation of pigment abundance and the ability of mutant tomato plants to adapt to high intensity light. The effects of high-intensity light were quantified in both high and low-pigment mutants, determining different parameters, like general leaf morphology, water content, content and composition of pigments, distribution of anthocyanins, photochemical activity and antioxidant capacity. In addition, the transcript levels of several genes involved in the processes of light signaling and in the biosynthesis of secondary metabolites were determined. In general, the article is carefully written and contains information that may prove important for people working in this field.

Authors speculate that an elevated expression of the HY5 gene may be linked to elevated flavonoid and carotenoid contents. It would be interesting to check if this can be reflected also in a mutant overexpressing this gene in a future study. Below are minor suggestions for the authors that may improve the readability the article.

Title: Authors may want to consider shortening the title, for example by omitting the words “high-pigment and low-pigment”.

Abstract: Please consider avoiding detailed numbering in this section. Please also avoid the use of abbreviations in the abstract and consider including those in the Introduction.

Introduction: Please avoid repetitions, like for example the information contained in the lines 35-36, that is already contained in the sentence below. Also the information in lines 50-52 that is repeated in lines 56-57. Some detailed information and relevant references on the genes related to the processes of light signaling and secondary metabolite biosynthesis should be included in this section.

Results: In order to facilitate comparisons by the reader, several results from the Tables could also be illustrated as graphs. Figure 3 might look more impressive if bars were colored. Please also consider modifying the titles of chapters 3.2, 3.4 and 3.5 giving more emphasis on the actual measurement, or the result, rather than the method used.

Discussion: Please consider further highlighting the relevance of the differences observed in the anthocyanin distribution in the different mutants.

Author Response

  1. Ashikhmin and colleagues present an article on the correlation of pigment abundance and the ability of mutant tomato plants to adapt to high intensity light. The effects of high-intensity light were quantified in both high and low-pigment mutants, determining different parameters, like general leaf morphology, water content, content and composition of pigments, distribution of anthocyanins, photochemical activity and antioxidant capacity. In addition, the transcript levels of several genes involved in the processes of light signaling and in the biosynthesis of secondary metabolites were determined. In general, the article is carefully written and contains information that may prove important for people working in this field.

Authors speculate that an elevated expression of the HY5 gene may be linked to elevated flavonoid and carotenoid contents. It would be interesting to check if this can be reflected also in a mutant overexpressing this gene in a future study

Answer: We sincerely appreciate the reviewer's positive assessment of our work and are grateful for the time invested in reviewing our manuscript.

  1. Title: Authors may want to consider shortening the title, for example by omitting the words “high-pigment and low-pigment”.

Answer: We agree with the improved title “The Adaptive Role of Carotenoids and Anthocyanins in Solanum lycopersicum Pigment Mutants Under High Irradiance”.

  1. Abstract: Please consider avoiding detailed numbering in this section. Please also avoid the use of abbreviations in the abstract and consider including those in the Introduction.

Answer: We agree.

Improved abstract «The effects of high-intensity light on the pigment content, photosynthetic rate, and fluorescence parameters of photosystem II in high pigment tomato mutants (referred to as 3005) and low pigment mutants (referred to as 3617) were investigated. This study also evaluated the dry weight percentage, antioxidant capacity of low molecular weight, expression patterns of primary light-dependent photoreceptor-regulated genes, and structural aspects of leaf mesophyll cells. The 3005 mutant displayed increased levels of photosynthetic pigments and anthocyanins, whereas the 3617 mutant demonstrated a heightened content of ultraviolet-absorbing pigments. The photosynthetic rate, photosystem II activity, antioxidant capacity, and carotenoid content were most pronounced in the high pigment mutant after prolonged exposure to intense light. This mutant also exhibited an increase in leaf thickness and water content when exposed to high-intensity light, suggesting superior physiological adaptability and reduced photoinhibition. Our findings indicate that the enhanced adaptability of the high pigment mutant might be attributed to increased flavonoid and carotenoid contents, leading to augmented expression of key genes associated with pigment synthesis and light regulation».

  1. Introduction: Please avoid repetitions, like for example the information contained in the lines 35-36, that is already contained in the sentence below. Also the information in lines 50-52 that is repeated in lines 56-57. Some detailed information and relevant references on the genes related to the processes of light signaling and secondary metabolite biosynthesis should be included in this section.

Answer: We agree. We improved the introduction extensively.

  1. Results: In order to facilitate comparisons by the reader, several results from the Tables could also be illustrated as graphs. Figure 3 might look more impressive if bars were colored. Please also consider modifying the titles of chapters 3.2, 3.4 and 3.5 giving more emphasis on the actual measurement, or the result, rather than the method used.

Answer: We agree. It is done. We concur with the reviewer's suggestion that graphs would provide a clearer representation. However, due to the sheer volume of data, creating them is difficult. To enhance the tables' readability, we implemented color coding in the header map. We trust this will address the reviewer's concerns and make the data more accessible.

  1. Discussion: Please consider further highlighting the relevance of the differences observed in the anthocyanin distribution in the different mutants.

Answer: We agree and added information on the anthocyanins.

The text:

We observed served high values of TEAC and elevated contents of anthocyanins and carotenoids in the 3005 mutant (Table 1, Figure 2). Elevated expression of light-regulated genes, such as CHS, PAL1 and FLS, also suggests an increase in flavonoid contents (Figure 3). It can be assumed that a high content of all these pigments contributes to the resistance of PA to long-term HIL

was replaced with new text:

We observed high values of TEAC and elevated contents of anthocyanins and carotenoids in the 3005 mutant (Table 1, Figure 2). In particular, a significant difference in pigments between 3005 and 3617 mutants was indicated at an intensity of 2000 µmol (photons) m-²s⁻¹. For example, at 72 h, the HIL anthocyanin content was higher in the 3005 mutant than in the 3617 mutant by almost 40 times and carotenoids by 1.5 times. A much higher content of anthocyanins under the above conditions was also observed in the parenchyma and epidermis of the 3005 mutant compared to the 3617 mutant. Elevated expression of light-regulated genes, such as CHS, PAL1 and FLS, also suggests an increase in flavonoid contents (Figure 3). It can be assumed that a high content of all these pigments contributes to the resistance of PA to long-term HIL.”

Round 2

Reviewer 1 Report

Comments and Suggestions for Authors

The authors have addressed all my comments and concerns, and have made revisions/corrections accordingly. This reviewer does not have further questions/comments.